# Defense Responses Stimulated by Bacillus subtilis NCD-2 Through Salicylate- and Jasmonate-Dependent Signaling Pathways Protect Cotton Against Verticillium Wilt

**DOI:** 10.3390/ijms26072987

**Published:** 2025-03-25

**Authors:** Shaojing Mo, Weisong Zhao, Yarui Wei, Zhenhe Su, Shezeng Li, Xiuyun Lu, Xiaoyun Zhang, Yuanhang Qu, Peipei Wang, Lihong Dong, Jiaqi Zhang, Qinggang Guo, Ping Ma

**Affiliations:** Key Laboratory of IPM on Crops in Northern Region of North China, Integrated Pest Management Innovation Centre of Hebei Province, Institute of Plant Protection, Hebei Academy of Agriculture and Forestry Sciences, Ministry of Agriculture and Rural Affairs of China, Baoding 071000, China; msjing1983@163.com (S.M.); zhaoweisong1985@163.com (W.Z.); weiyr2017@163.com (Y.W.); suzhenhe0202@126.com (Z.S.); shezengli@163.com (S.L.); luxiuyun03@163.com (X.L.); zxy_zxl@163.com (X.Z.); quyuanhang1990@163.com (Y.Q.); wangpeipei0010@163.com (P.W.); lihong_dong56@126.com (L.D.); jiaqi_321@yeah.net (J.Z.)

**Keywords:** *Bacillus subtilis* NCD-2, cotton Verticillium wilt, biological control, induced resistance, signaling pathway

## Abstract

*Bacillus subtilis* NCD-2 demonstrates exceptional biocontrol potential against cotton Verticillium wilt. While previous studies have established its direct antifungal activity (e.g., inhibiting *Verticillium dahliae* mycelial growth and spore germination), our work reveals a novel mechanism: NCD-2 primes systemic resistance in cotton by activating plant immune-signaling pathways. Firstly, transcriptional profiling uncovered that NCD-2 triggers a defense response in roots analogous to *V. dahliae* infection, allowing cotton to maintain a more balanced state when confronted with pathogen attacks. Meanwhile, the mutant strains ∆fen and ∆srf—defective in lipopeptide synthesis—also improved cotton resistance to Verticillium wilt by activating partially identical defense pathways in cotton roots. Furthermore, the application of lipopeptide compounds derived from NCD-2, particularly surfactin and fengycin, could enhance host resistance to *V. dahliae*. Using an RT-qPCR approach, we found that numerous resistance-related genes were induced by these lipopeptide compounds. The up-regulation of SA/JA pathway markers (e.g., *NPR1*, *ICS1*, *COI1*, and *LOX1*) revealed NCD-2’s activation of plant immune signaling. Using virus-induced gene silencing (VIGS), we conclusively linked SA and JA signaling to NCD-2-induced defense priming. Silencing either pathway abolished resistance, highlighting their indispensable coordination. By bridging mechanistic insights and agricultural applicability, our work positions NCD-2 as a sustainable alternative to conventional fungicides, addressing both crop productivity and environmental health.

## 1. Introduction

Upland cotton (*Gossypium hirsutum* L.), a major source of fibers and oilseeds, is one of the most important economic crops globally. Verticillium wilt, caused by *Verticillium dahliae*, is the most devastating disease in agricultural production and has the broadest host ranges among plant fungal pathogens [1]. This soil-borne pathogen can reach high incidences and cause tremendous yield losses in cotton, resulting in approximately 2.5 million hectares of Verticillium-wilt-infected cotton fields and direct economic losses reaching USD 250–310 million annually in China [2,3]. Due to the devastating damage it causes in cotton production, cotton Verticillium wilt is also referred to as “the cotton cancer” [4]. However, the pathogen’s stable dormant structure, microsclerotia, long-term variability, and co-evolution with the host plant make it extremely difficult to control [5,6,7,8]. Due to the poor transportation of conventional fungicides to the xylem vessels, chemical control is virtually ineffective against cotton Verticillium wilt. Although soil fumigation can be effective, it is prohibitively expensive for field production and harmful to the environment, making the control of cotton Verticillium wilt a major challenge [9].

The use of biological control for cotton Verticillium wilt has recently garnered considerable interest due to its environmentally friendly nature [10,11,12,13]. The *Bacillus* genus, being ubiquitous in the environment, has been widely implemented in biological control as a probiotic. Numerous studies have addressed the potential use of the *Bacillus* genus for the biocontrol of cotton Verticillium wilt [12,14,15,16]. Among the identified *Bacillus* species, *B. subtilis* is the most utilized and studied, making it a highly promising candidate for agricultural applications [17,18]. *B. subtilis*, being a Gram-positive and nonpathogenic bacterium, is widely employed to study the biological characteristics of the *Bacillus* genus, such as secondary metabolite production and biofilm development [19]. *B. subtilis* protects plants from pathogens through a variety of direct and indirect methods, including competing with phytopathogens for nutrients and spatial sites, producing antimicrobials, and triggering induced systemic resistance (ISR) [17,20]. This last method has emerged as a crucial mechanism through which bacteria and fungi in the rhizosphere prepare the entire plant for enhanced defense against a broad range of pathogens [21,22].

To cope with pathogen challenges, plants employ pattern-triggered immunity (PTI) and effector-triggered immunity (ETI) to discern and confront pathogen infections. PTI is initiated through the recognition of pathogen-associated molecular patterns, while ETI is activated by the recognition of pathogen effectors [23]. The onset of PTI and ETI often triggers induced resistance in tissues distanced from the infection site, a phenomenon known as systemic acquired resistance (SAR) [24,25]. The accumulation and signaling of the plant hormone salicylic acid (SA) are critical for the establishment of SAR [26]. Besides SA, the plant hormones jasmonic acid (JA) and ethylene (ET) have also been proven to regulate the plant immune system [27,28]. Beneficial rhizobacteria can produce various secondary metabolites that induce plant systemic resistance, and these are called elicitors. Numerous studies have demonstrated that volatile organic compounds such as 2R,3R-butanediol and acetoin—as well as lipopeptides (LPs) like surfactin and fengycin produced by *Bacillus* strains—elicit ISR [29,30,31,32]. Alongside these elicitors, bacillomycin D, bacillaene, macrolactin, difficidin, bacilysin, and exopolysaccharides produced by *Bacillus amyloliquefaciens* SQR9 also contribute to ISR [33]. The strong correlation between defense-inducing activity and the presence of surfactin suggests that this compound plays a central role in stimulating the plant immune system [34]. Similar to pathogen-induced SAR, the sensing of elicitors triggers several plant signaling pathways to stimulate an immune response.

*Bacillus subtilis* NCD-2, previously isolated from the cotton rhizosphere in our laboratory, can considerably control cotton Verticillium wilt in field trials [14]. Because of its excellent effectiveness against various soil-borne diseases, strain NCD-2 has been developed as a biofungicide [35]. Multiple studies have analyzed the factors influencing the colonization characteristics, the key compounds with antimicrobial activity, and the effects on the microbial community structure in the rhizosphere of the NCD-2 strain [20,35,36,37,38,39]. Accumulated evidence has confirmed that activation of the plant immune system by beneficial microbes plays a crucial role in suppressing plant pathogens [40,41]. However, little is known about the defense response induced by *Bacillus subtilis* NCD-2 in cotton and its role in fighting Verticillium wilt.

In this study, we aimed to address the following two critical research questions: (1) How does *Bacillus subtilis* NCD-2 induce systemic resistance in cotton against Verticillium wilt at the molecular level? (2) Which specific phytohormone signaling pathways are involved in this biocontrol process? To systematically investigate these questions, we first demonstrated that both strain NCD-2 and its lipopeptide compounds could significantly enhance cotton resistance to Verticillium wilt. Using integrated transcriptome profiling and RT-qPCR validation, we then deciphered the key molecular mechanisms underlying this induced resistance. Furthermore, through targeted silencing of core genes in plant hormone signaling pathways, we specifically confirmed the essential roles of SA and JA signaling in NCD-2-mediated defense responses. Our findings not only elucidate the molecular basis of biocontrol efficacy by *B. subtilis* NCD-2 but also provide new insights into the crosstalk between microbial elicitors and plant immune pathways in combating soil-borne diseases.

## 2. Results

### 2.1. B. subtilis NCD-2 Confers Cotton Verticillium Wilt Resistance

Preliminary research in our laboratory found that the *B. subtilis* strain NCD-2 can effectively control cotton Verticillium wilt. Our laboratory also conducted a continuing field experiment, which demonstrated that the biocontrol efficacy of strain NCD-2 against cotton Verticillium wilt can reach as high as 90% [14].

In this study, we evaluated the control effect of strain NCD-2 against Verticillium wilt in a greenhouse setting. One week after root irrigation with either strain NCD-2 or a control (LB medium), the cotton seedlings were inoculated with a highly virulent *V. dahliae* strain, wx-1. Approximately 3 weeks post-infection, the plants pre-treated with strain NCD-2 exhibited significantly decreased disease severity as evidenced by less wilting and a lower disease index compared to the control plants (Figure 1A,B). Additionally, the fungal recovery from the stem sections of strain NCD-2-pre-treated plants was markedly lower than that from control plants (Figure 1C). Similarly, the detection of fungal biomass followed the same trend (Figure 1D). Thus, root irrigation with strain NCD-2 effectively suppresses cotton Verticillium wilt under our greenhouse conditions.

We have previously found that the lipopeptide compounds produced by strain NCD-2 can significantly inhibit the growth of pathogenic fungi in vitro [35]. Surprisingly, treatment with strain NCD-2 did not result in a decrease in the populations of *V. dahliae* in the cotton rhizosphere [38]. In this study, we devised experiments to investigate whether strain NCD-2 can trigger a defense response in cotton against Verticillium wilt. After being pre-treated with strain NCD-2 for 7 days, the cotton roots were thoroughly rinsed with distilled water. Thereafter, the cotton seedlings were inoculated with *V. dahliae* using the root-dip method and transferred to new plastic cups filled with sterile substrate. Two weeks later, we initiated a regular log of the severity of cotton disease. Simultaneously, we used two single-structural mutants, ∆FenC (now ∆fen) and ∆srfAA (now ∆srf), as well as a double mutant ΔSrfAA/FenC (now ∆sf)—which formed altered lipopeptide compounds [39]—for the cotton pre-treatment. Plants treated with strain NCD-2 showed a significant reduction in disease symptoms, which included fewer incidences of leaf chlorosis, less wilting, and a lower disease index compared to the control plants (Figure 1E,F). In the meantime, seedlings pre-treated with ∆fen and ∆srf also demonstrated improved resistance to Verticillium wilt compared to the control seedlings (Figure 1E,F). Conversely, the double mutant *∆sf* could not confer resistance to cotton Verticillium wilt (Figure 1E and Appendix A). These results suggest that, in addition to its antagonistic effect on *V. dahliae*, strain NCD-2 may enhance the tolerance of cotton to Verticillium wilt by activating the defense system.

### 2.2. NCD-2-Activated Cotton Defense Response Against V. dahliae

To further understand the resistance mechanisms of strain NCD-2-treated cotton toward *V. dahliae*, we conducted a transcriptome analysis using roots and leaves from wx-1-inoculated cotton seedlings, pre-treated and not pre-treated with strain NCD-2. Specifically, the treatments were divided into control (CK), strain NCD-2 only (NCD), *V. dahliae* only (VD), and strain NCD-2 plus *V. dahliae* (NVD).

Initially, we analyzed the differentially expressed genes (DEGs) in the leaf tissues. The NCD-2 treatment led to 1908 DEGs in the leaves, which included 552 up-regulated genes and 1356 down-regulated genes. In contrast, we found 2837 DEGs in the leaves treated solely with *V. dahliae*; these comprised 1570 up-regulated genes and 1267 down-regulated genes (Appendix A).

Gene ontology (GO) analysis revealed categories such as regulation of the SA biosynthetic process, immune system process, response to external biotic stimulus, and defense response to other organisms, enriched among the DEGs resulting from NCD-2 treatment (Appendix A). Moreover, Kyoto Encyclopedia of Genes and Genomes (KEGG) analysis indicated that pathways including plant–pathogen interactions, the phosphatidylinositol signaling system, MAPK signaling pathway, and phenylpropanoid biosynthesis were enriched among NCD-2-treatment-induced DEGs. This implies the activation of the host immune response by NCD-2 treatment (Appendix A).

In terms of DEGs induced by the VD treatment, categories like response to chitin, response to hydrogen peroxide, SAR, and response to reactive oxygen species were visible in the GO analysis (Appendix A). At the same time, pathways including plant–pathogen interaction, MAPK signaling pathway, plant hormone signal transduction, and phenylpropanoid biosynthesis were enriched in the KEGG analysis (Appendix A). Furthermore, the analysis showed 546 genes overlapped between the NCD-2 and VD treatments (Appendix A), accounting for only 19% of the DEGs resulting from the VD treatment.

Next, we analyzed the DEGs from root tissues. Both GO and KEGG analyses yielded similar enrichment results for the NCD-2 and VD treatments. The GO analysis indicated that, of the most significantly enriched categories, 12 out of 20 overlapped between the NCD-2 and VD treatments (Figure 2A). Meanwhile, the KEGG analysis demonstrated that the number of overlapped pathways was 14 (Figure 2B). These data suggest a high similarity of DEGs in the NCD-2- and VD-treated cotton roots. Indeed, the Venn diagram indicated the number of overlapping DEGs to be 1021, which accounted for a substantial 56.4% of the DEGs resulting from the VD treatment (Figure 2C). In addition, these genes exhibited the same changes in expression patterns across both treatments (Figure 2D). Moreover, we performed a GO enrichment analysis on DEGs that we screened using more stringent criteria (Padjust < 0.05). The findings revealed that the enriched GO terms were primarily associated with secondary metabolite biosynthesis and metabolism, such as the diterpenoid biosynthetic process, terpenoid biosynthetic process, and terpene metabolic process. This indicates the induction of defense responses in the NCD-2 treated roots (Appendix A).

To gain a deeper understanding of NCD-2’s role in modifying the defense response of the host cotton, we compared the number of DEGs between the VD/CK and NVD/NCD groups. As illustrated in Figure 2E, when contrasted with the uninfected *V. dahliae* control, we discovered 1810 DEGs in the *V. dahliae*-infected roots (VD/CK). However, when pre-treated with NCD-2, the number of DEGs dramatically decreased to 968. These findings suggest that NCD-2-pre-treated cotton might trigger a less extensive response to *V. dahliae* infection compared to untreated cotton. In terms of energy metabolism, this result indicates that NCD-2-pre-treated cotton maintains a more stable state when faced with pathogen stress compared to untreated cotton. However, the conditions in leaf tissue were contrasting. The number of DEGs in the NVD/NCD and VD/CK groups were 4343 and 2837, respectively (Figure 2E).

In addition to transcriptome data, we further confirmed the defense-inducing effect of strain NCD-2 by detecting the activities of two defense-related enzymes. Results showed that, 7 days after strain NCD-2 treatment, the cotton leaves exhibited significantly increased SOD and POD enzyme activity (Figure 3A,B). Consistent with the enzyme activity data, we also witnessed elevated levels of H_2_O_2_ in the strain NCD-2-treated leaves (Figure 3C).

### 2.3. Two NCD-2 Mutants Defected in Lipopeptide Synthesis Result in Partially Identical Activation Effects in Cotton Plants

Lipopeptide compounds have been reported to be the major elicitors in inducing plant systemic resistance [33]. In this study, we investigated the roles of surfactin and fengycin, two principal lipopeptide compounds produced by strain NCD-2, in the activation of defense response in cotton plants. One week after root irrigation with wild-type (NCD), mutant strain (∆fen and ∆srf), or a blank control (CK), we collected cotton root tissues for transcriptome analysis. Surprisingly, not the wild-type but the ∆srf treatment resulted in the greatest number of DEGs in cotton roots. We identified 2106, 4035, and 2962 DEGs in the NCD-2-, ∆srf-, and ∆fen-pre-treated cotton roots, respectively (Appendix A). The numbers of DEGs in each group compared are shown in a Venn diagram (Figure 4A). Contrary to our expectation, neither ∆fen- nor ∆srf-induced DEGs were completely included in the wild-type induced DEGs. The enrichment of the MAPK signaling pathway, plant hormone signal transduction, and phenylpropanoid biosynthesis in the KEGG analysis suggests that ∆fen and ∆srf mutants activated defense-related signaling pathways in cotton roots (Figure 4B). We then focused our analysis on the ∆fen- and ∆srf-induced DEGs. DEGs specifically identified in ∆srf-pre-treated cotton were 2312, while the number dropped to 1239 in ∆fen-treated cotton (Figure 4A). We observed overlaps of five pathway items in the two KEGG analyses, including plant hormone signal transduction, MAPK signaling pathway, phenylpropanoid biosynthesis, glucosinolate biosynthesis, and zeatin biosynthesis (Figure 4C). There were 1723 DEGs present in ∆fen- and ∆srf-treated cotton roots (Figure 4A). Further analyses revealed that these DEGs displayed the same expression pattern in both treatments. Specifically, 654 genes were up-regulated by either ∆fen or ∆srf treatment and 1068 genes were down-regulated by either treatment (Figure 4D). All things considered, we can conclude that the two mutants, ∆fen and ∆srf, mediate the activation of partially identical defense pathways in cotton roots. However, the roles of surfactin and fengycin in the activation of defense response in cotton plants were not clarified.

### 2.4. NCD-2-Derived Lipopeptide Compounds Activated Cotton Defense Response Against V. dahliae

The transcriptome analysis results suggest that strain NCD-2 might activate the defense responses in cotton plants, aiding in resistance against *V. dahliae* invasion. To assess whether the lipopeptide compounds produced by strain NCD-2 could effectively activate the defense response in cotton plants, we examined the effect of the cell-free culture filtrate (CF) on cotton after *V. dahliae* infection. The CF was derived from a strain NCD-2 culture with Landy broth, mainly used for lipopeptide production [42]. Cotton treated with Landy broth and *V. dahliae* was used as a control, and we marked it as Vd. To avoid direct antagonistic effects, the CF and culture medium were applied and rinsed off before the introduction of the pathogen. The development of disease symptoms was significantly reduced in CF-treated plants compared to the control plants (Figure 5A). Further investigation of the disease index also confirmed this result (Figure 5B).

To elucidate the molecular mechanism underlying the resistance of strain NCD-2-derived lipopeptide compounds to *V. dahliae*, we examined the expression pattern of the resistance-related genes reported in the published literature. We compiled as many resistance-related genes tied to cotton Verticillium wilt resistance as we could. Genes sourced from *Gossypium barbadense* were re-identified from upland cotton, with PCR primers designed according to the gene sequence of the upland cotton. We listed the 43 resistance-related genes’ information in Appendix A. Then, we used RT-qPCR analysis to measure the relative expression of these 43 genes between CF and CK (Landy broth) treatments. Among these genes, 9 and 5 displayed “undetermined” results in roots and leaves, respectively, due to low expression levels (Appendix A). In root tissue, genes such as *GhLAC15*, *GhVe1*, *GhPAO*, and *GhGPA* were significantly up-regulated by CF treatment (Figure 5C) and are known to positively regulate cotton resistance to *V. dahliae* [43,44,45,46]. However, many pathogenesis-related (*PR*) genes, including commonly regarded marker genes *PR1* and *PR5*, were significantly down-regulated by CF treatment (Figure 5D). This suggests that while CF treatment effectively activates cotton plants’ defense response, it does not up-regulate *PR* genes in cotton root tissue.

### 2.5. NCD-2-Derived Surfactin- and Fengycin-Activated Cotton Defense Response Against V. dahliae

To determine whether the lipopeptide compounds, surfactin and fengycin, derived from strain NCD-2 increase cotton disease resistance, we isolated surfactin and fengycin from the lipopeptide extract of strain NCD-2 using fast protein liquid chromatography (FPLC). Referring to the chromatographic graphs from the fengycin and surfactin standards, the solutes collected were those at the same retention time (Figure 6A,B). We then concentrated these extracts, dissolved them in methanol, and established their concentrations according to the standard curves (Appendix A). Afterward, we further tested the fengycin and surfactin from strain NCD-2 for their potential to induce defense activity in cotton plants. To avoid direct antagonistic effects, the compounds were applied and then washed off before the inoculation of the pathogen. Disease reduction, rated at 25 days post-infection (Figure 6C), revealed that fengycin and surfactin efficiently protected cotton plants, achieving an average reduction in disease incidence of 38.3% and 28.0%, respectively (Figure 6D).

The fact that there is a strong correlation between defense-inducing activity and the amount of surfactin production has previously proven that surfactin is a primary compound that stimulates plant immune-related responses in *B. subtilis*/*amyloliquefaciens* [34]. Our focus here is on the defense-inducing activity of fengycin, separated from strain NCD-2, on the cotton root tissue. We utilized RT-qPCR analysis to review the expression pattern of resistance-related genes (refer to Appendix A). In line with the cell-free CF treatment, fengycin effectively sparked the defense response in cotton roots, leading to a significant up-regulation of a range of resistance-related genes, including *GhMLP28*, *GhACIF1*, *GhGLP2*, *GhDSC1*, *GhCYP86A1*, *GhSOBIR1*, *GhSARD*, and *GhTSA1* (Figure 6E). These genes were all reported to positively regulate cotton’s resistance to *V. dahliae* [47,48,49,50,51,52,53,54]. Take *GhMLP28,* for example, the expression of the *GhMLP28* gene was prompted by *V. dahliae* inoculation, and the knockdown of *GhMLP28* expression by virus-induced gene silencing resulted in increased susceptibility of cotton plants to *V. dahliae* infection [48]. These findings imply a direct stimulation of the cotton defense system by strain NCD-2-derived fengycin.

### 2.6. Both SA and JA Pathways Are Required for NCD-2-Mediated Activation of Plant Defense Response

The plant hormones SA and JA are both important regulators of the plant immune system. Usually, the activation of multiple resistance pathways, when a plant faces pathogen infections, depends on the SA and JA signaling pathways. Meanwhile, studies have demonstrated that JA and ET are key players in regulating PGPR-mediated ISR in many scenarios [21]. To examine whether the SA or JA signaling pathways were involved in the strain NCD-2-mediated defense response in cotton plants, we evaluated the transcription level of the SA and JA signaling genes in leaves and roots. For SA signaling, marker genes involved in either SA biosynthesis or signal transduction, including *NPR1*, *ICSI*, *PAL1*, and *EDSI*, displayed increased transcript levels in the leaf and root tissues after CF treatment (Figure 7A and Appendix A). Simultaneously, the transcription of the JA-inducible marker genes *COI1*, *LOX1*, *AOS*, and *OPR3* increased significantly in the leaf tissue (Figure 7B). The situation was similar in the root tissue, except for *LOX1*, which was not activated (Appendix A). These data suggest that the SA and JA signaling pathways in cotton plants may be activated by CF produced by strain NCD-2.

To further confirm the necessity of the strain NCD-2-mediated defense response in SA and JA signaling pathways, we utilized virus-induced gene silencing (VIGS) to silence genes that have primary roles in these two pathways. We created VIGS constructs of *GhICS1*, *GhNPR1*, and *GhCOI1*, which are key genes in the SA and JA signaling pathways (Appendix A), and infiltrated these into cotton seedlings using the Agrobacterium tumefaciens strain GV3101 as a vector. For optimal silencing effect, we simultaneously silenced *GhICS1* and *GhNPR1* by infiltrating with equal volumes of vectors, designated as *TRV:NI*. We used *TRV:00* as a blank control and *TRV:GhSSI2* as a positive control to verify the effectiveness of the gene-silencing system. As anticipated, cotton leaves presented a noticeable lesion mimic and curled phenotype 2 weeks following agroinfiltration with *TRV:GhSSI2* (Figure 8A). This outcome confirmed that the VIGS system functioned effectively under our experimental conditions. Meanwhile, we observed no significant phenotypic differences between the *TRV:NI*, *TRV:COI1*, and *TRV:00* plants (Figure 8B). The substantial reduction in the transcripts of target genes in the VIGS plants indicates that *GhICS1*, *GhNPR1*, and *GhCOI1* were efficiently silenced in the cotton plants (Figure 8C). Confirming the blockage of the SA and JA signaling pathways involved applying exogenous SA and MeJA to the corresponding VIGS cotton plants through foliar spraying. We then used RT-qPCR to assess the expression level of marker genes linked with SA and JA signaling. For SA signaling, there was no variation in the *PR1* gene, which markedly increased in *TRV:00* during SA treatment in the *TRV:NI* (Appendix A). With regards to JA signaling, *PDF1.2* was not enhanced by MeJA in *TRV:COI1* as it was in the *TRV:00* plants (Appendix A). These findings suggest that signaling pathways were effectively blocked in the *TRV:NI* and *TRV:COI1* plants.

We subsequently investigated the transcription level of the *PR* genes during CF treatment in the VIGS plants. The results revealed that the transcripts of *PR1*, *PR2*, *PR3*, and *PR5* were considerably elevated by CF in the *TRV:00* plants (Figure 8D,E). However, the *TRV:NI* plants exhibited non-elevated *PR1* and a relatively lower *PR5* induction due to CF treatment (Figure 8D). Similarly, the *TRV:COI1* plants indicated non-elevated *PR3* and down-regulated *PR2* resulting from CF treatment (Figure 8E).

Furthermore, we examined the phenotypic changes in disease resistance to Verticillium wilt in the *TRV:NI* and *TRV:COI1* plants. As shown in Figure 9A, when treated with strain NCD-2, the control plants significantly enhanced Verticillium wilt resistance but this was not the case in either the *TRV:NI* or *TRV:COI1* plants. Consistent with this, the corresponding disease index and the stem dissection, showing a darker brown vascular bundle (Figure 9B,C), both supported the idea that silencing the SA and JA pathways decreased strain NCD-2-induced cotton tolerance to Verticillium wilt. In addition, we noted a significant loss in the growth promotion effect on the silenced cotton plants (Appendix A). In summary, the results indicated that the SA and JA signaling pathways are essential for NCD-2-induced defense activation in cotton plants.

## 3. Discussion

Biological control for cotton Verticillium wilt is considered an environmentally friendly strategy. The ability of *Bacillus* species to biosynthesize specific compounds with antimicrobial activity has caused widespread use as biocontrol agents [55]. Cotton seedlings treated with *B. subtilis* EBS03, *B. amyloliquefaciens* YZU-SG146, and *B. halotolerans* Y6 [15,16,56] all indicated suppressed Verticillium wilt. The potential to produce over two dozen antimicrobial compounds (AMCs) and a genome devoting at least 4–5% to antibiotic production make *B. subtilis* group strains ideal biocontrol agents [55,57]. *B. subtilis* strain NCD-2, previously isolated by our laboratory, stands out from the *Bacillus* species reported to have effects on Verticillium wilt and acts as the main component for the first registered biofungicide in China [35]. The superior antifungal effect and beneficial influences on the host plant microbiome of this strain have already been detailed in our prior work [35,38]. In this study, we demonstrate for the first time that treatment of cotton plants with strain NCD-2 could effectively induce immune defense in cotton, thereby enhancing Verticillium wilt resistance (Figure 1). This result is not surprising, given the findings that concentrations of antifungal LPs detected in plants were meager [58,59]. Additionally, despite possessing an extensive genetic arsenal for the formation of bioactive compounds, the *Bacillus* strains only express a small portion of antibiotics in the plant rhizosphere, excluding the antibacterial polyketides [60]. Combined with the fact that, despite showing no antifungal activity toward pathogen *F. oxysporum*, *B. cereus* strain EC9 could effectively protect Kalanchoe against root rot disease [40], we speculate that the induction of the host immune defense acts as a critical—and perhaps a primary factor—in suppressing plant pathogens by some *Bacilli*. The significant reduction in disease symptoms and the lower disease index in the NCD-2, ∆fen, ∆srf, and lipopeptide-compound pre-treated cotton compared with the control plants confirms our speculation (Figure 1, Figure 4 and Figure 5). Notably, compromised biocontrol efficacy was observed in the Δsf mutant treatment. This phenomenon may stem from the predominant production of fengycin and surfactin lipopeptides by the NCD-2 strain, with other lipopeptide classes being synthesized at suboptimal concentrations insufficient to elicit immune responses in cotton plants. Furthermore, studies indicate that fengycin contributes critically to rhizospheric colonization of strain NCD-2. The Δsf double mutation likely caused a dual reduction: (1) a diminished root colonization capacity of NCD-2; and (2) attenuated accumulation of lipopeptide elicitors. This combined deficiency ultimately compromised the strain’s ability to enhance Verticillium wilt resistance in cotton [39].

So far, the majority of studies looking into the mechanism for using beneficial microorganisms to control cotton Verticillium wilt have focused on antagonism [11,15,61,62]. In our study, transcriptome sequencing technology was used to analyze the defense mechanism of strain NCD-2 against *Verticillium dahliae* in cotton. We found that strain NCD-2 treatment activated the host immune response in root and leaf tissues. When comparing these DEGs against the wx-1-inoculated cotton, we discovered that a similar defense activation was triggered in the roots by the beneficial microbe strain, whereas the DEGs in the leaf tissue were significantly different from the pathogen treatment (Figure 2 and Appendix A).

A recent study into the defense mechanism of the endophytic fungus *Chaetomium globosum* CEF-082 against cotton Verticillium wilt revealed that pre-inoculation with CEF-082 strengthened the defense response of cotton plants after *V. dahliae* treatment [63]. Our study examined the tripartite interactions of the pathogenic fungus–plant–beneficial microbe from a different perspective. Compared to untreated cotton, pre-treatment with strain NCD-2 resulted in a less extensive reaction to *V. dahliae* infection in the root tissue. In other words, NCD-2-pre-treated cotton remained in a more homeostatic state when facing pathogen stress compared to untreated cotton. Recent studies on endophytic *Streptomyces hygroscopicus* OsiSh-2 mediated disease resistance to rice blast and barley root endophyte *Serendipita vermifera* (*Sv*) against the soil-borne pathogen *Bipolaris sorokiniana* (*Bs*) support our findings [64,65]. These studies found that the number of DEPs in rice dropped from 1022 to 512 following inoculation with the OsiSh-2 strain [64]. They also demonstrated that the number of DEGs in barley was reduced from 2743 to 1517 following simultaneous inoculation with *Sv* and beneficial bacteria, pointing to the mobilization of host transcription in barley [65].

Contrary to the root condition, the reaction in the leaf tissue was more intense after treatment with the NCD-2 strain. *V. dahliae* led to a stronger response in the leaves than the non-inoculated NCD-2 treatment, and the subsequent enrichment of GO terms related to energy metabolism (Figure 2E and Appendix A). We speculate that when facing the invasion of pathogens from the root tissue, the leaves armed with ISR need to adjust their transcriptional programming strategy and enhance energy metabolism to assist the roots in combating pathogens.

*Bacillus* spp. can produce multiple elicitors to stimulate plant defense responses, such as surfactin, fengycin, 2-aminobenzoic acid, 2,3-butanediol, acetoin, and dimethyl disulfide [31,66,67,68]. Additionally, subtilisin secreted by *Bacillus velezensis* LJ02 can induce SAR in *Nicotiana benthamiana* and enhance the plant’s resistance to *Botrytis cinerea* [69]. Among these, surfactin is considered the most essential elicitor, which stimulates plant immune responses through actions on the lipid bilayer of the plasma membrane [34,70,71]. Additionally, pure fengycins confer notable *Botrytis cinerea* resistance to bean plants similar to the fengycin-producing strain S499 [31].

In this study, we separated surfactin and fengycin from the lipopeptide extract of the strain NCD-2 strain using FPLC. We then assessed the defence-inducing activity of these two compounds through a resistance experiment in the *V. dahliae*/cotton pathosystem. We found that surfactin and fengycin from strain NCD-2 significantly enhanced the cotton’s resistance to Verticillium wilt. Using nine mutants deficient in elicitor production, we found that *Bacillus amyloliquefaciens* SQR9 elicitors such as fengycin, bacillomycin D, surfactin, bacillaene, and macrolactin could induce different plant defense genes [33]. We attempted to establish whether surfactin and fengycin from strain NCD-2 induce the activation of different defense genes in cotton. Transcriptome data showed that ∆srf resulted in 4035 DEGs, among which 2312 were specifically identified in ∆srf treatment but not in ∆fen treatment. Simultaneously, ∆fen resulted in 2962 DEGs, among which 1239 were specifically identified in ∆fen but not in ∆srf treatment. Also, there were 1723 DEGs present in ∆fen- and ∆srf-pre-treated cotton (Figure 3). Given these data, we hypothesized that the 2312 DEGs were activated by fengycins and the 1239 DEGs by surfactin. Meanwhile, the 1723 DEGs might be activated by other elicitors present in strain NCD-2 (Appendix A).

ISR is a mechanism by which plant growth-promoting rhizobacteria (PGPR) in the rhizosphere activate the entire plant for enhanced defense against pathogens [72]. This important mechanism was first reported in 1991 by three research groups, respectively [73,74,75]. Considering that PGPR could induce resistance in distal parts of the plant separate from the inducer, hundreds of studies have leveraged this ability to protect dicot and monocot plants against diseases, especially foliar diseases occurring in the above-ground parts [76,77,78,79,80,81]. However, cotton Verticillium wilt is a soil-borne disease that involves a pathogenic fungus invading the cotton vascular system from the root tissue, eventually causing a systemic infection [82].

To reveal the molecular mechanisms underlying NCD-2-stimulated cotton defense responses, we centered our study on the resistance-related genes involved in cotton Verticillium wilt resistance in the root tissue. For this purpose, we collected as many resistance-related genes as possible and detected their expression levels using RT-qPCR. According to Figure 5C and Figure 6E, multiple resistance-related genes—including laccase gene *GhLAC15*, leucine-rich repeat-receptor-like protein *GhVe1*, major latex protein *GhMLP28*, Avr9/Cf-9-INDUCED F-BOX1 *GhACIF1*, and germin-like protein *GhGLP2*—were up-regulated by the CF and fengycin treatments. Simultaneously, we examined the expression level of *PR* genes and found that *PR1*, *PR3*, *PR5*, *PR6*, and *PR10* were significantly down-regulated by the CF treatment. These results align with the facts that inoculation with the beneficial microbe *Stenotrophomonas rhizophila* SR80 for soil-borne *Fusarium pseudograminearum* suppression tended to reduce the expression of PR genes in wheat shoots and that *Pseudomonas fluorescens* WCS417r-ISR protected radish and *Arabidopsis* against *Fusarium oxysporum* without accumulating PR proteins [41,83,84].

Rhizobacteria-mediated ISR was initially thought to be independent of SA but predominantly dependent on the JA- and/or ET-signaling pathways [85,86,87,88,89]. However, numerous beneficial microbes have also been reported to trigger an SA-dependent type of ISR, similar to pathogen-induced SAR. For instance, it was demonstrated that *PaeniBacillus alvei* K165 enhances *V. dahliae* resistance in *Arabidopsis thaliana* by activating the SA signaling pathway and increasing the expression of PR-1, PR-2, and PR-5 genes [90]. Numerous studies have found that beneficial rhizobacteria may trigger ISR by simultaneously activating the SA- and JA/ET-signaling pathways [80,91]. It is worth noting that beneficial rhizobacteria not only defend against foliar diseases but also trigger plant hormone signaling pathways to defend against soil-borne diseases [41,92].

A CF treatment increased transcript levels of marker genes involved in SA and JA biosynthesis or signal transduction, suggesting SA and JA signaling pathways in cotton plants are activated by strain NCD-2 (Figure 6). In addition to using the RT-qPCR method to detect the expression levels of marker genes, genotypes impaired in SA accumulation, JA biosynthesis, and ET production in *Arabidopsis*, tomato, and maize are typically employed to clarify the involvement of hormone signaling pathways in the PGPR-mediated ISR [92,93,94].

Unlike model plants, acquiring mutant plants for the allotetraploid upland cotton is very challenging. VIGS is considered a relatively simple and efficient approach to down-regulating target genes and has become an important tool in cotton gene function research [51,95,96]. In our current study, we successfully knocked down the expression of marker genes in the SA and JA signaling pathways, respectively. The *TRV:NI* and *TRV:COI1* plants were unable to perceive the exogenous application of SA and MeJA, confirming the successful blocking of the SA and JA signaling pathways (Appendix A). Further, when treated with strain NCD-2, the control plants significantly enhanced Verticillium wilt resistance, while the SA- and JA-signaling-pathway-impaired plants—*TRV:NI* and *TRV:COI1*—both lost NCD-2-induced tolerance to *V. dahliae*. Utilizing these plant materials, we present strong evidence that SA and JA signaling pathways are crucial for NCD-2-induced defense activation in cotton plants. This study supports the use of VIGS as a novel strategy to examine PGPR-mediated defense mechanisms in cotton and other non-model plant species.

## 4. Materials and Methods

### 4.1. Growth Conditions of Plants

The cotton (cv. Ejing 1, susceptible to Verticillium wilt) seeds were disinfected with 75% (*v*/*v*) ethanol and 2.5% (*v*/*v*) NaClO, then washed with distilled water at least 3 times. The seeds were then planted in sterilized soil or vermiculite and grown in a greenhouse or an automatically controlled environment chamber for different experiments. The greenhouse and growth chamber offered the growth conditions of a 16-h light/8-h dark cycle under 30 ± 2 °C/25 ± 2 °C (day/night) temperature and a 16-h light/8-h dark cycle under a temperature of 24 °C, respectively.

### 4.2. Preparation of Bacterial Suspensions and Fungal Inoculum

*B. subtilis* strain NCD-2 and its fengycin/surfactin-deficient mutant (∆fen and ∆srf) and double mutant (∆sf) were stored at −80 °C in our laboratory. Strain descriptions: The Δfen strain is a *fenC* deletion mutant of NCD-2 and is deficient in fengycin synthetase. The Δsrf strain is an *srfAA* knockout mutant of NCD-2 and is deficient in surfactin biosynthesis. The Δsf strain is a double mutant derived from NCD-2, lacking both *fenC* and *srfAA* [35,39].

After being activated on LB solid medium, the strains were then grown at 37 °C on LB liquid medium with shaking at 180 rpm. For lipopeptide production, strain NCD-2 was grown in Landy broth [42] at 30 °C and 180 rpm.

The *V. dahliae* strain wx-1 was isolated and preserved in our laboratory. The fungus was cultured on PDA agar medium at 25 °C for 7 days. Mycelial plugs were transferred into liquid potato dextrose medium for 5 d at 25 °C until the spore concentration reached ~10^8^ spores/mL. After filtering through four layers of sterile gauze, the spore suspension was centrifuged at 5000× *g* for 10 min and adjusted to a final concentration of 1 × 10^7^ conidia/mL with sterile distilled water for inoculation.

### 4.3. Separation and Concentration Detection of Lipopeptides by FPLC

Lipopeptide compounds were extracted using the method described by Su [20]. In brief, strain NCD-2 was cultured in Landy broth [42] at 30 °C for 3 days on a shaker. The culture was centrifuged at 4 °C and 8000× *g* for 30 min, then the supernatant was collected and adjusted to pH 2.0 with 6 mol/L HCl. The mixture was stored at 4 °C for 12 h and centrifuged at 10,000× *g* for 20 min. Then, the extracts were collected and dissolved in 10 mL methanol. After passing through 0.45-micrometer filters, the crude lipopeptides were obtained. The crude lipopeptides were separated and purified according to previous descriptions [20]. In brief, the crude lipopeptides were purified using an AKTA Purifier system (GE Healthcare, Uppsala, Sweden) equipped with a SOURCE 5RPC ST 4.6/150 column. The separation was achieved through a binary solvent system consisting of the following: solvent A, 2% acetonitrile containing 0.065% trifluoroacetic acid (TFA) (*v*/*v*); solvent B, 80% acetonitrile containing 0.05% TFA (*v*/*v*). Chromatographic separation was performed using a linear acetonitrile gradient from 0% to 100% of solvent B over 57 min at a flow rate of 1 mL/min. The detection wavelength was 215 nm. All major chromatographic peaks were automatically fractionated by the FPLC system. The peaks representing surfactin and fengycin were collected and the collections were concentrated using a rotary evaporator. The content of surfactin and fengycin was calculated according to the standard curve, which was made from commercial surfactin and fengycin purchased from GLPBIO and MedChemExpress.

In this study, we defined the centrifuged culture that was prepared for surfactin and fengycin separation as cell-free culture filtrate (CF) and this was used for further research.

### 4.4. Protection Efficacy of Bacterial Suspensions, CF, Fengycins, and Surfactin Against Cotton V. dahliae and Sample Collection

Cotton seedlings planted in the sterilized soil for 2 weeks were treated with a 5-milliliter cell suspension of the NCD-2 strain at an OD600 of 1.0 or with 5 mL of LB medium (control) by soil drenching. After 7 days of irrigation, the plant roots were damaged from the bottom of the plastic container and either inoculated with the *V. dahliae* spore suspension (1 × 10^7^ spores mL^−1^) or treated with water. This treatment protocol was employed to investigate the Verticillium-wilt-resistance phenotype in cotton.

Seedlings planted in the sterilized vermiculite were treated with cell suspensions of NCD-2, *∆fen*, *∆srf*, and *∆sf* strains or LB medium (as the control) as in the method above. After one week, the plants were uprooted and the roots were rinsed thoroughly with distilled water, followed by pathogen inoculation using a root-dipping method. The inoculated plants were replanted on the newly changed substrate. This treatment protocol simultaneously facilitated the following: (1) the systematic investigation of Verticillium-wilt-resistance phenotypes; and (2) the standardized collection of samples for subsequent transcriptome profiling. For the sample collection, the following was conducted: (1) One week after root irrigation with the strain NCD-2 wild-type (NCD), the mutant strain (*∆fen* and *∆srf*), or the blank control (CK), cotton root tissues were collected for transcriptome analysis. (2) One week after root irrigation with wild-type NCD-2 or the control solution, plants were inoculated with *V. dahliae* spores. Leaf and root samples were collected 2 days post-inoculation. The experiment comprised four treatments: CK (control, no treatment); NCD (NCD-2 treatment only); VD (*V. dahliae* inoculation only); NVD (a combined NCD-2 pre-treatment and *V. dahliae* challenge).

For the CF, fengycin, and surfactin treatments, cotton seedlings planted in the sterilized vermiculite for 1 week were transferred to plastic pots (diameter: 10 cm) that contained 200 mL of 1/2 Hoagland solution. After 1 week of hydroponics, the CF, fengycins, and surfactin were added and reached a final concentration of 5% (*v*/*v*), 5 μM, and 5 μM, respectively. The controls were added with the corresponding volume of Landy broth and methanol. Two days later, the roots were rinsed thoroughly with distilled water, followed by pathogen inoculation using a root-dipping method. The disease index was scored about 3 weeks after inoculation, according to previous descriptions [97]. At the same time, leaf and root tissues were sampled at 1 day post-CF, after fengycin and surfactin treatment, for further RT-qPCR study.

### 4.5. Fungal Recovery Assay and Biomass Quantification

For the fungal recovery assay, corresponding parts of the stems were cut off from the plants and surface sterilized. Then, the stem sections were cultured on PDA medium and incubated at 25 °C. The colonies of *V. dahliae* were observed after four days post-culture. For the fungal biomass quantification, total DNA was obtained from the first true leaves of cotton using a Plant Genomic DNA Kit (TIANGEN, China). RT-qPCR was performed using cotton- and fungal-specific primers (GhHIS3 and Vd-ITS) on the extracted DNA. Fungal biomass was estimated by calculating the ratio between the fungal (*ITS*) and cotton (*HIS3*) genes.

### 4.6. Quantitative Real-Time Reverse Transcription–PCR

Cotton tissues (0.05 g) were collected for total RNA extraction by using an RNAprep Pure Plant Plus kit (TIANGEN, Beijing, China). The resulting RNA was reverse-transcribed to cDNA using the TransScript One-Step gDNA Removal and cDNA Synthesis SuperMix (TransGen Biotech, Beijing, China). The genes’ expressions were analyzed by RT-qPCR using the ABI QuantStudio 6 Flex system (Applied Biosystems, Foster City, CA, USA). The primers used for RT-qPCR are listed in Appendix A. The GhHIS3 was used as an internal reference. Three biological replicates were used for each analysis with three technical replicates each.

### 4.7. VIGS Procedure

Cotton VIGS was performed according to methods described previously [96]. The conserved regions of *GhNPR1*, *GhICS1*, and *GhCOI1* were individually selected as targets for VIGS and synthesized by gene-synthesis technology (GENEWIZ, Suzhou, China). The obtained fragments were then constructed into pTRV2 [98]. The infiltrated plants were then grown in the automatically controlled environment chamber at 24 °C under a 16/8h light/dark cycle with 60% humidity. Two weeks after infiltration, RNA was extracted from the cotton leaves to detect the expression of target genes. The *TRV:SSI2* construct was used as a positive marker for evaluating VIGS efficiency.

### 4.8. Application of Salicylic Acid, Methyl Jasmonate, and CF Treatments to the VIGS Plants

The cotton seedlings that effectively silenced the *NPR1* and *ICS1* genes were foliar sprayed with salicylic acid (500 μM) and methanol (control), while the *COI1*-silenced plants were sprayed with methyl jasmonate (100 μM) and methanol (control). The *TRV:00* plants were sprayed with salicylic acid, methyl jasmonate, and methanol. Leaf samples were collected at 6 h after treatment and immediately frozen in liquid nitrogen. For the CF treatment, the cotton seedlings were transferred to hydroponics culture for 2 weeks after VIGS infiltration. Then, the CF was added and reached a final concentration of 5% (*v*/*v*). Leaf samples were collected at 24 h after treatment and immediately frozen in liquid nitrogen.

### 4.9. Measurement of H_2_O_2_ and Activities of Defense-Related Enzymes

Cotton leaves were sampled at 7 days post-NCD-2 treatment, and 0.1g of detached leaf tissue was fully ground in liquid to determine the activities of defense-related enzymes using SOD and POD test kits (Beijing Solarbio Science & Technology Co., Ltd., Beijing, China) according to the manufacturer’s instructions, respectively. The level of H_2_O_2_ was determined according to the manufacturer’s method with an H_2_O_2_ assay kit (Nanjing Jiancheng Bioengineering Institute, Nanjing, China).

### 4.10. Transcriptome Sequencing

The total RNA was extracted using an RNAprep Pure Plant Plus kit (TIANGEN, Beijing, China) according to the manufacturer’s instructions. RNA sequencing and data analysis were performed by the Shanghai Majorbio company. The differentially expressed genes were defined using the DESeq R package with the negative binomial distribution (FDR < 0.05). The DEGs used for the analysis are under the criteria of the corrected *p*-value < 0.05 and absolute log2 ratio ≥ 1.

## 5. Conclusions

This study elucidates the dual-mode biocontrol mechanism of *Bacillus subtilis* NCD-2 against cotton Verticillium wilt, demonstrating that this strain effectively controls the disease not only through its previously identified strong antimicrobial activity but also via activation of systemic resistance in cotton plants. Key findings include the following: (1) NCD-2 enhances cotton’s resistance to Verticillium wilt by triggering defense responses and allowing cotton roots to maintain a more balanced state when confronting pathogen attacks. (2) Mutant strains ∆fen and ∆srf—which are deficient in fengycins and surfactin—also improve cotton’s resistance to Verticillium wilt and mediate the activation of partially identical defense pathways in the plant. (3) Fengycin derived from NCD-2 boosts host resistance to *V. dahliae* by inducing the expression of numerous resistance-related genes. (4) Blocking the SA and JA signaling pathways through VIGS technology confirms that both signaling pathways are essential for NCD-2-induced defense activation. Together, the results of this study reveal a new mechanism through which the *Bacillus subtilis* NCD-2 participates in the adjustment of plant resistance via SA and JA signals.

## Figures and Tables

**Figure 1 ijms-26-02987-f001:**
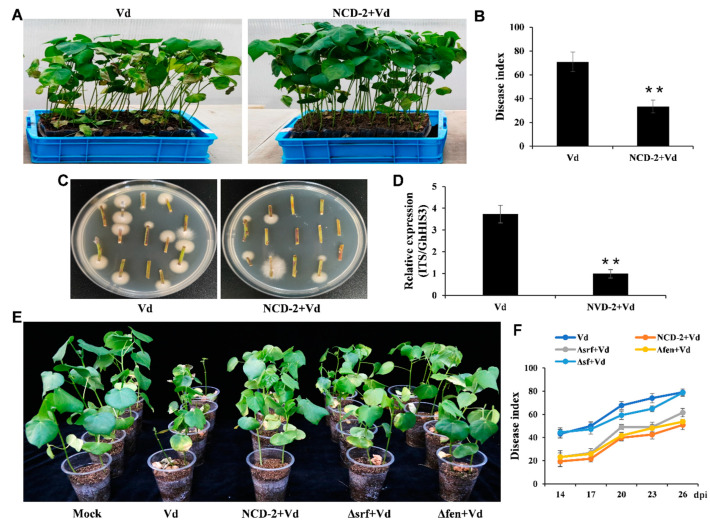
*B. subtilis* NCD-2-pre-treated cotton plants display increased resistance to *V. dahliae*: (**A**) Disease symptoms of NCD-2 suspension pre-treated and CK plants after inoculation with *V. dahliae* strain wx-1. Photographs were taken 25 days post-wx-1 inoculation. (**B**) The disease index of the NCD-2-pre-treated plants and CK plants. The values are the means ± SE, n = 3 (** *p* < 0.01, Student’s *t*-test). (**C**) Fungal recovery experiments showed decreased fungal growth in NCD-2-pre-treated plants. Photographs were taken at 4 d post-recovery. (**D**) Fungal biomass detection of the infected cotton plants. The fungal biomass was represented by the expression of the fungal *ITS* gene compared with cotton *GhHIS3* gene. (**E**) Disease symptoms of NCD-2-, ∆fen-, ∆srf-, and CK-pre-treated plants after inoculation with *V. dahliae* strain wx-1. Photographs were taken 25 days post-wx-1 inoculation. (**F**) Disease indices of NCD-2-, ∆fen-, ∆srf-, ∆sf-, and CK-pre-treated plants were determined after inoculation with wx-1 at 14, 17, 20, 23, and 26 d. The values are the means ± SE, n = 3.

**Figure 2 ijms-26-02987-f002:**
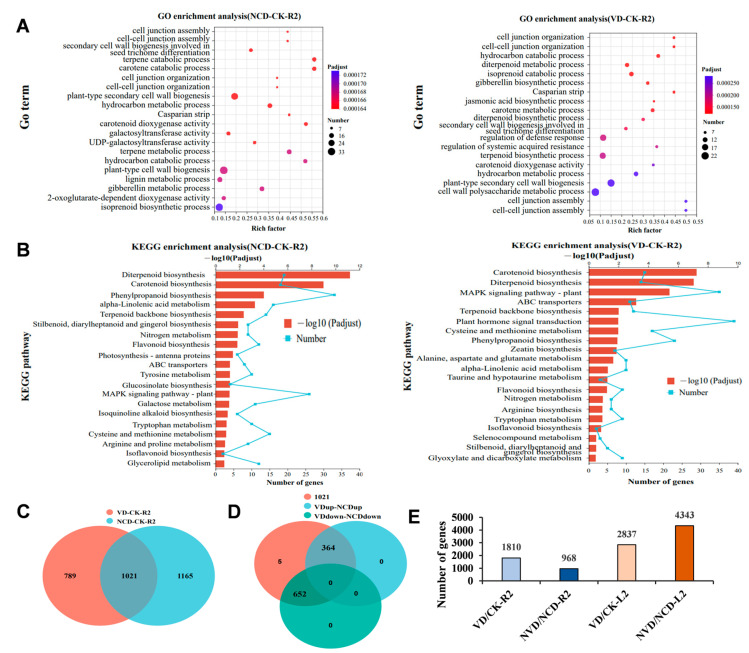
*B. subtilis* NCD-2 treatment activates a defense response in the cotton roots similar to that of *V. dahliae* treatment: (**A**) GO enrichment analysis of the differentially expressed genes (DEGs) in NCD-2- or *V. dahliae*-treated plants. (**B**) KEGG pathway enrichment analysis of the DEGs in NCD-2- or *V. dahliae*-treated plants. (**C**) The overlap of DEG abundance between the NCD-2 and *V. dahliae* treatments is presented in a Venn diagram. (**D**) Venn diagram shows the expression pattern of the 1021 DEGs in the NCD-2 and *V. dahliae* treatments. (**E**) Numbers of DEGs in each compared group.

**Figure 3 ijms-26-02987-f003:**
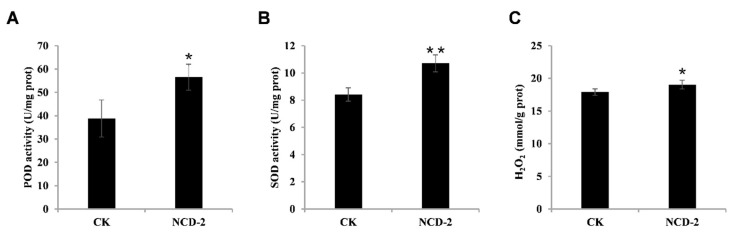
*B. subtilis* NCD-2 treatment activates elevated ROS levels in the cotton plants: (**A**) POD activity; (**B**) SOD activity; (**C**) H_2_O_2_ content. Values represent the means ± SE for three biological replicates (* *p* < 0.05, ** *p* < 0.01, Student’s *t*-test).

**Figure 4 ijms-26-02987-f004:**
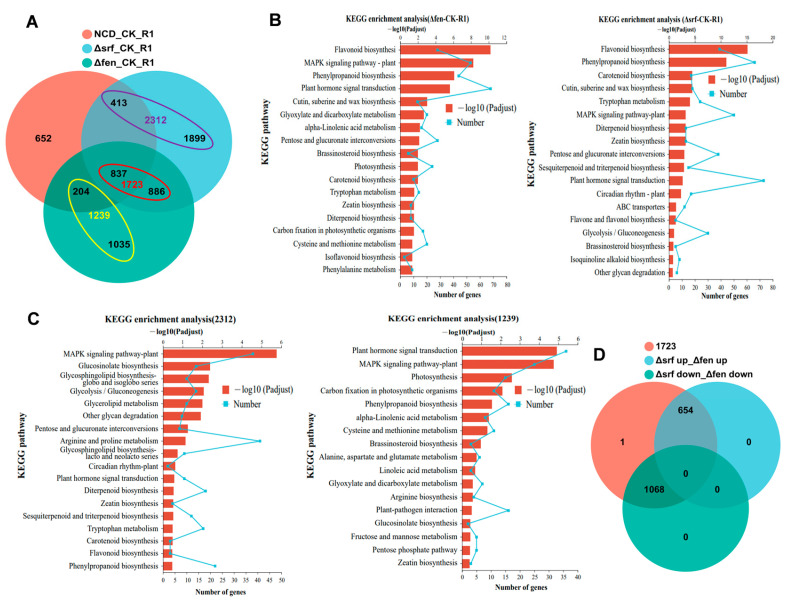
*B. subtilis* mutants Δsrf and Δfen activate partially identical defense responses in the cotton roots: (**A**) The overlap of DEG abundance between the NCD-2, Δsrf, and Δfen treatments is presented in a Venn diagram. (**B**) KEGG pathway enrichment analysis of the DEGs in Δsrf- or Δfen-treated plants. (**C**) KEGG pathway enrichment analysis of the 2312 DEGs and 1239 DEGs. (**D**) Venn diagram showed the expression pattern of the 1723 DEGs in the Δsrf and Δfen treatments.

**Figure 5 ijms-26-02987-f005:**
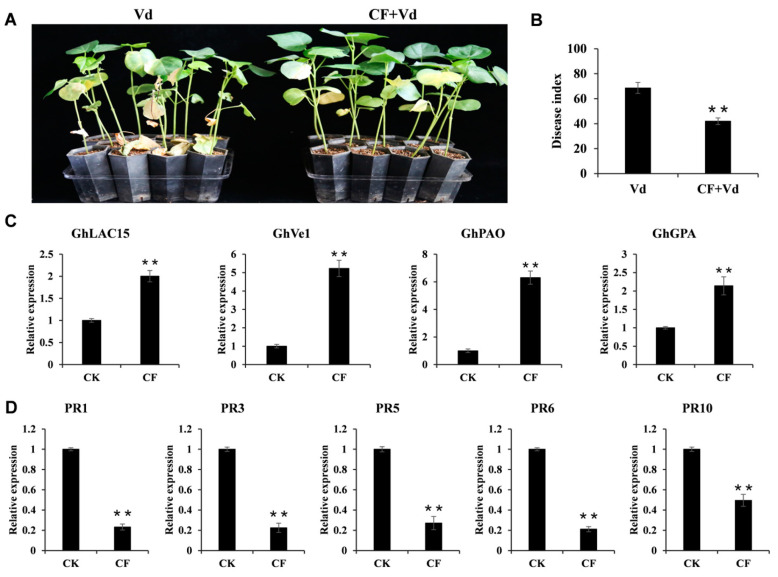
Lipopeptides produced by *B. subtilis* NCD-2 activate the resistance-related gene and enhance cotton resistance to *V. dahliae*: (**A**) Disease symptoms of cell-free culture filtrate (CF) pre-treated and CK plants after inoculation with *V. dahliae* strain wx-1. Photographs were taken 25 days post-wx-1 inoculation. (**B**) The disease index of the CF-pre-treated plants and CK plants. The values are the means ± SE, n = 3 (** *p* < 0.01, Student’s *t*-test). (**C**) RT-qPCR analysis of resistance-related gene expression in the cotton root upon CF and CK treatment. *GhHIS3* was used as an internal control. Values represent the means ± SE for three biological replicates (** *p* < 0.01, Student’s *t*-test). (**D**) RT-qPCR analysis of *PR* gene expression in the cotton root upon CF and CK treatment. Values represent the means ± SE for three biological replicates (** *p* < 0.01, Student’s *t*-test).

**Figure 6 ijms-26-02987-f006:**
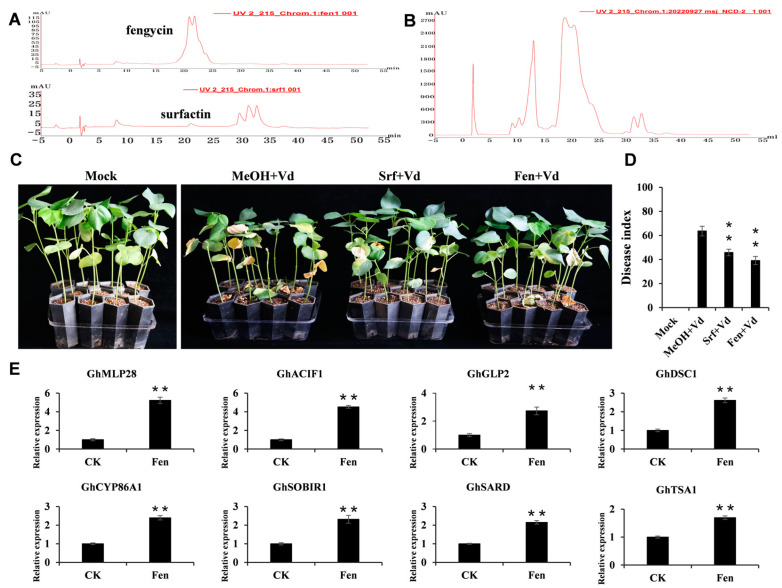
Fengycin produced by *B. subtilis* NCD-2 promotes the resistance-related gene and enhances cotton resistance to *V. dahliae*: (**A**) FPLC analysis of the reference standards surfactin and fengycin. (**B**) FPLC analysis of the lipopeptides produced by strain NCD-2. Surfactin and fengycins were collected according to the retention time. (**C**) Disease symptoms of surfactin- and fengycin-pre-treated plants and CK plants after inoculation with *V. dahliae* strain wx-1. Photographs were taken 25 days post-wx-1 inoculation. (**D**) The disease index of the surfactin- and fengycin-pre-treated plants and CK plants. The values are the means ± SE, n = 3 (** *p* < 0.01, Student’s *t*-test). (**E**) RT-qPCR analysis of resistance-related gene expression in the cotton root upon fengycin and CK treatment. The values are the means ± SE for three biological replicates (** *p* < 0.01, Student’s *t*-test).

**Figure 7 ijms-26-02987-f007:**
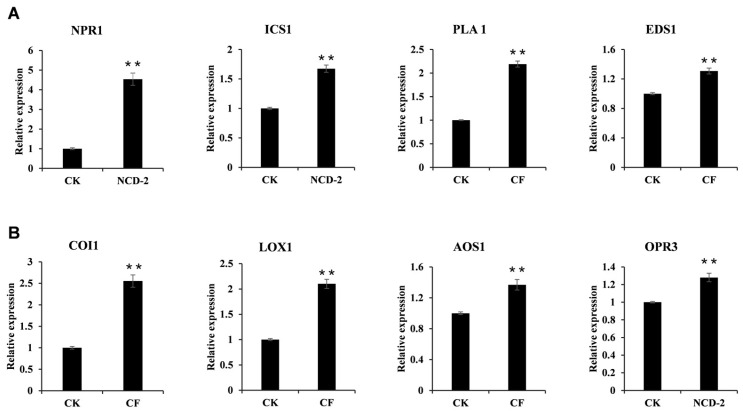
Lipopeptides produced by *B. subtilis* NCD-2 activates the SA and JA biosynthesis and signaling pathways in cotton: (**A**) Expression levels of SA signaling pathway genes in the cotton leaves upon CF and CK treatments. The values are the means ± SE for three biological replicates (** *p* < 0.01, Student’s *t*-test). (**B**) The transcript levels of JA signal transduction pathway genes in the cotton leaves upon CF and CK treatment. The values are the means ± SE for three biological replicates (** *p* < 0.01, Student’s *t*-test).

**Figure 8 ijms-26-02987-f008:**
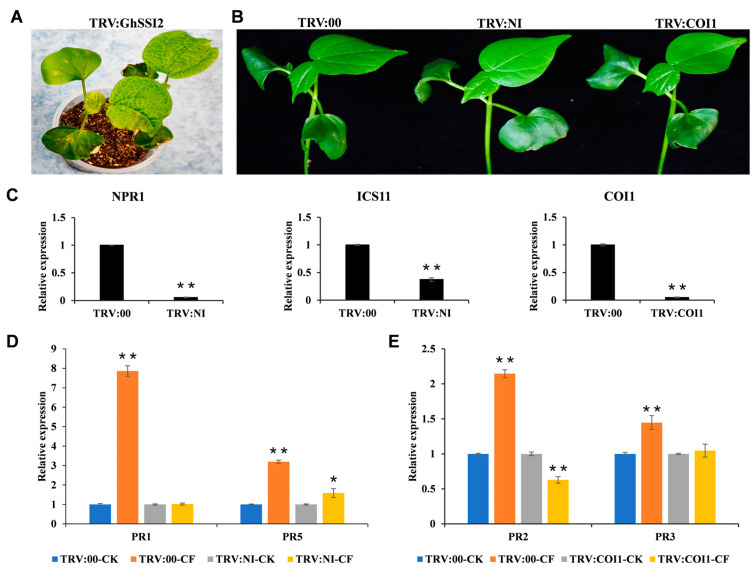
Suppression of SA and JA signaling pathways block the NCD-2-mediated systemic defense response in cotton: (**A**) Spontaneous lesion phenotype of the plants inoculated with *TRV:SSI2* for 15 d. (**B**) Morphological features of *TRV:NI* (*TRV:NPR1*&*ICS1*) and *TRV:COI1* plants showed no difference with *TRV:00* plants. Photograph was obtained 15 d after infiltration. (**C**) Verification of *GhNPR1*, *GhICS1*, and *GhCOI1* silencing by RT-qPCR in the corresponding VIGS plants. (**D**,**E**) *TRV:NI* and *TRV:COI* plants impaired in the inducement of PR gene expression by the CF treatment. Values represent the means ± SE for three biological replicates (* *p* < 0.05, ** *p* < 0.01, Student’s *t*-test).

**Figure 9 ijms-26-02987-f009:**
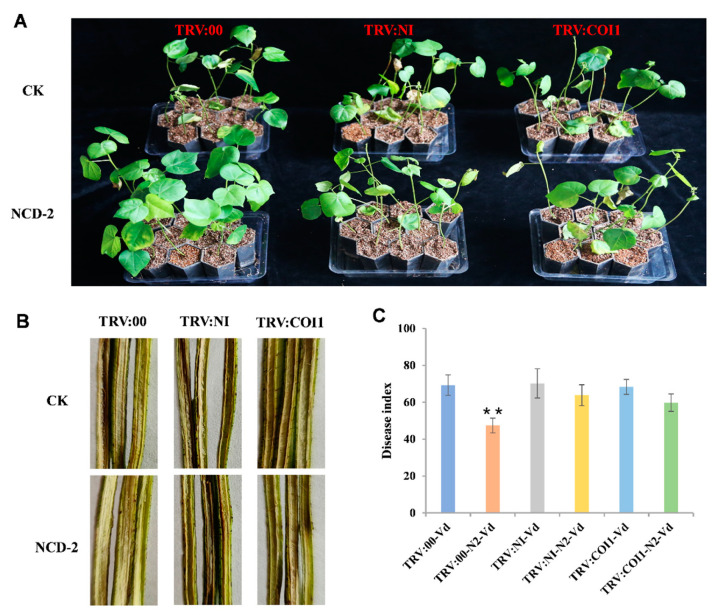
SA and JA signaling pathways are essential for NCD-2-induced resistance to Verticillium wilt. (**A**) Disease symptoms of the VIGS plants pre-treated with NCD-2 or CK after inoculation with *V. dahliae*. Photographs were taken 21 days post-*V. dahliae* infection. (**B**) Longitudinal sections of the cotton stems at 21 d after *V. dahliae* inoculation. (**C**) The disease index of the corresponding VIGS plants. Values represent the means ± SE for three biological replicates (** *p* < 0.01, Student’s *t*-test).

## Data Availability

All data and materials are included in the article and the Appendix A. Raw data for RNA-seq have been submitted to the National Center for Biotechnology Information with accession number: PRJNA1201903.

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
