# Peer review of "Defense Responses Stimulated by Bacillus subtilis NCD-2 Through Salicylate- and Jasmonate-Dependent Signaling Pathways Protect Cotton Against Verticillium Wilt"

_ijms, 2025, doi:10.3390/ijms26072987_

Round 1

Reviewer 1 Report

Comments and Suggestions for Authors

I reviewed the ms, it has some interests in the field production, and provides many data to explain the role of NCD-2 resistant against Verticillium Wilt infection. I have some concerns should be addressed before acceptance.

  1. In 2.1, you indicate that NCD-2 against Verticillium Wilt in field has been published, but you perform it again in the greenhouse. If there are difference between these two sites. Please explain.
  2. In all the figures, they are not clear, please improve the resolution. Please indicate more information about Δfen and Δsrf. In fig1a,e, the phenotype of Vd, NCD2-Vd were not consistent with each other in 25 days post wx-1 inoculation.
  3. In fig3, I suggest to measure the proline and MDA content.
  4. In fig5C, D, there are no SD in the Vd line, please check.
  5. In fig5, please unify the physiology index or gene expression measurement. Such as that in fig6d, it should indicate the disease index of Mock, MeOH+Vd, Srf+Vd, Fen+Vd.
  6. In fig7, I suggest to determine the content of JA and SA.
  7. If the SA and JA could response to Verticillium Wilt infection.
  8. In fig9D, the NI and COI1 did not response to Verticillium Wilt infection, please explain.

Reviewer 2 Report

Comments and Suggestions for Authors

This research is logically clear and rich in data, but it needs further improvement in methodological details, result interpretation, and writing standardization.

Abstract: The innovative points need to be more prominent, and the practical application value can be supplemented in the conclusion part.

Introduction: It is recommended to clarify the research question in the last paragraph.

Materials and Methods:

  1. The information on how to obtain the two mutant materials is missing for me.
  2. The specific parameters of FPLC need to be supplemented in the lipopeptide separation step.
  3. The description of experimental groups is unclear: for example, abbreviations such as "NCD-2 only (NCD), V. dahliae only (VD)" need to be defined in advance in the methods.
  4. The past tense should be used consistently in the methods section, and "offer" in line 539 needs to be revised.

Results and Discussion:

  1. It is unclear why the author did not include Δsf in Figure 1E. The fact that Δsf cannot confer resistance to Verticillium wilt in cotton is not explained in the main text (Does the double mutation lead to a complete loss of lipopeptides? Supplementary discussion is required).
  2. Figure 5D shows the downregulation of PR genes, but the relationship with the SA pathway is not discussed (Is it possible that the SA signal is not fully activated? Analysis is required in the discussion).
  3. Hormone content was not measured, and only gene expression was relied on for indirect speculation.

Other suggestions:

  1. Add a scale bar to the phenotype figures.
  2. Supplement data on lipopeptide yield and verification of lipopeptide purification.
  3. Polish the language to reduce colloquial expressions, such as "Everyone knows" in line 326.

Comments on the Quality of English Language
  1. Polish the language to reduce colloquial expressions, such as "Everyone knows" in line 326.

Reviewer 3 Report

Comments and Suggestions for Authors

The main aim of the present MS is to clarify the background of the efficient use of Bacillus subtilis NCD-2 as biological control against cot-11 ton Verticillium wilt.

The protective effect was previously demonstrated, here the proven fact was further investigated at gene expression level, thus  DEG analyses was performed.

The MS is demanding, easy to follow and understand. All the required information is found in the introduction. The experimental setup is enough complex to provide a detailed conclusion on the observed results.

Changes were investigated after root irrigation with NCD-2, in order to confirm the action mechanism of lipopeptide compounds of NCD-2  two different approaches were applied, first two single-structural mutants, ∆FenC (now ∆fen), ∆srfAA (now ∆srf), and a double mutant ΔSrfAA/FenC (now ∆sf), were also used, and after then cell-free culture filtrate of NCD-2 was also tested.

Question: Have you got any information about besides the lipopeptide compounds what other compounds could be present in the filtrate, and do they have any putative roles in the induction of resistance of cotton?

In a third approach, they isolated surfactin and fengycin, and monitored their effect during infection.

Interesting both the application of mutants and the treatment with individual lipopeptides resulted in dedicated differences. Thus this complex and logical experimental setup showed very important and novel results.

They also stated "This data suggests that the SA and JA signaling pathways in cotton plants may be activated by CF produced by strain NCD-2. "

Question again: does the filtrate contain any other compound which is capable to induce plnat hormone signalling? Please clarify and highlight more what are the most important and responsible components that induce these extensive genetic arsenal of the defence mechanism after the treatment with CF!

In a last step, virus-induced gene silencing was used to silence genes that have roles in SA and JA signalling pathways.

Please rewrite the Conclusion to be more informative and attention focusing.
